

# Woodlice change the habitat use of spiders in a different food chain

Stefanie M. Guiliano[1,*], Cerina M. Karr[1,*], Nathalie R. Sommer[2] and Robert W. Buchkowski[2,3]

[1] Northeastern University, Boston, MA, United States of America
[2] School of Forestry & Environmental Studies, Yale University, New Haven, CT, United States of America
[3] Department of Biology, University of Western Ontario, London, Canada
[*] These authors contributed equally to this work.

## ABSTRACT

**Background**. In old field systems, the common woodlouse may have an indirect effect on a nursery web spider. Woodlice and nursery web spiders feed in different food chains, yet previous work demonstrated that the presence of woodlice is correlated with higher predation success by nursery web spiders upon their grasshopper prey. This finding suggested a new hypothesis which links two seemingly disparate food chains: when woodlice are present, the spider predator or the grasshopper prey changes their location in the vegetative canopy in a way that increases their spatial overlap and therefore predation rate. However, warming temperatures may complicate this phenomenon. The spider cannot tolerate thermal stress, meaning warming temperatures may cause the spider to move downwards in the vegetative canopy or otherwise alter its response to woodlice. Therefore, we would expect warming and woodlice presence to have an interactive effect on predation rate.

**Methods**. We conducted behavioral experiments in 2015, 2017, and 2018 to track habitat domains—the use of the vegetative canopy space by grasshoppers and spiders—in experimental cages. Then, we used three models of spider movement to try to explain the response of spiders to woodlice: expected net energy gain, signal detection theory, and individual-based modelling.

**Results**. Habitat domain observations revealed that spiders shift upward in the canopy when woodlice are present, but the corresponding effect on grasshopper prey survival was variable over the different years of study. Under warming conditions, spiders remained lower in the canopy regardless of the presence of woodlice, suggesting that thermal stress is more important than the effect of woodlice. Our modelling results suggest that spiders do not need to move away from woodlice to maximize net energy gain (expected net energy gain and signal detection theory models). Instead spider behavior is consistent with the null hypothesis that they move away from unsuccessful encounters with woodlice (individual-based simulation). We conclude that mapping how predator behavior changes across biotic (e.g. woodlouse presence) *and* abiotic conditions (e.g. temperature) may be critical to anticipate changes in ecosystem dynamics.

Corresponding author
Robert W. Buchkowski,
robert.buchkowski@gmail.com

## INTRODUCTION

Cross food chain interactions can occur when species share some portion of their habitat. These interactions are often mediated by changes in behavior or habitat-use, so-called trait-mediated interactions (*Ohgushi & Schmitz, 2012*; *Buchkowski & Schmitz, 2015*). Consumer species that acquire their energy from primary production or decomposition are said to feed in the plant-based and detritus-based food chains, respectively. They are prime candidates for cross food chain interactions because they can have overlapping habitat domains (*Bardgett & Wardle, 2010*; *Zou et al., 2016*; *Northfield, Barton & Schmitz, 2017*). However, the data required to understand behavioral interactions between animals in separate food chains are rarely collected (*Schmitz, 2006*; *Zhao et al., 2013*).

Preliminary data suggests that cross food chain interactions might have an important effect on trophic dynamics in New England old fields. The sit-and-wait predator *Pisaurina mira* (nursery web spider) typically has a positive indirect effect on plant diversity and soil nitrogen, because it causes its herbivore prey *Melanoplus femurrubrum* (red-legged grasshopper) to switch from feeding on grasses to feeding on the dominant goldenrod species (*Schmitz, 2006*). When grasshoppers depress goldenrod biomass, it reduces the amount of nitrogen being removed from the soil and so increases nitrogen mineralization. The effect on soil nitrogen occurs because grasshoppers remain at a high density and change their foraging decisions under the risk of predation. However, when the detritivorous woodlouse *Oniscus asellus* is present, grasshopper survival decreases and the effect of *P. mira* on soil nitrogen disappears (*Buchkowski & Schmitz, 2015*). One hypothesis explaining this phenomenon is that woodlice increase *P. mira* predation rate by changing how often spiders and grasshoppers interact in the old field canopy. In other words, woodlice shift *P. mira* from a predator with primarily trait-mediated effects to one with density-mediated effects (*Schmitz, 2006*).

The exact mechanism explaining the spider, grasshopper, and woodlouse interaction remains unclear. Either spiders or grasshoppers could shift their position within the canopy in the presence of woodlice, leading to a higher encounter rate and higher predation rate. Grasshoppers might shift in response to a risk from woodlice, which are opportunistically predaceous (*Edney, Allen & McFarlane, 1974*; *Le Clec'h et al., 2013*). Spiders might shift in response to woodlice because woodlice are not an accessible prey item and their movement distracts spiders from true prey items (i.e., signal detection theory; *Green & Swets, 1966*; *Staddon & Gendron, 1983*; *Getty & Krebs, 1985*; *Abbott & Sherratt, 2013*). Spiders do not eat woodlice, because feeding on woodlice requires significant morphological and biochemical adaptations that this species does not possess (*Vizueta et al., 2019*). We have no evidence that woodlice pose a risk to grasshoppers, nor that spiders would be attracted towards woodlice as a potential prey item (we observed no predation in shared terraria). So, we hypothesized that spiders move upwards in the canopy to avoid distractions caused by woodlice movement, and thereby come into closer contact with grasshoppers. Behavioral observations of grasshopper and spider habitat domain—their respective use of the canopy space—are necessary to test this hypothesis.
Interactions between spiders, grasshoppers, and woodlice may also be mediated by abiotic stressors. High temperatures cause spiders to move downwards in the canopy, away from grasshoppers (*Barton & Schmitz, 2009*). Grasshoppers in old-field ecosystems do not change their location in the canopy in response to warming, because they can tolerate higher temperatures than spiders (*Barton & Schmitz, 2009*). This raises the question of whether any effect of woodlice on grasshopper survival would hold under climate warming. If spiders respond both to woodlice (by moving up) and climate warming (by moving down) the consequences for grasshopper survival should be null.

We conducted a series of behavioral experiments to evaluate spider and grasshopper habitat domains under woodlouse presence and warming. Habitat domains use the mean spatial location to measure where animals spend their time, and use the variance to determine how much animals move (*Miller, Ament & Schmitz, 2014*; *Rosenblatt, Wyatt & Schmitz, 2019*). Habitat domain is a useful metric for our study because it can be used to predict differences in predation rate (*Northfield, Barton & Schmitz, 2017*). We used the habitat domains from our behavioral observations to calculate spider attack rates for our theoretical models.

We used three theoretical models to explain the response of spiders and grasshoppers to woodlouse presence. The first two models were competing models that consider the net energy gain of spiders as they attack grasshoppers and woodlice (*Abbott & Sherratt, 2013*). The third model was an individual-based model that tests whether woodlice encounters can cause spiders to occupy a different canopy position.

We documented the habitat domains of grasshoppers and spiders in the presence and absence of woodlice in cage experiments in 2015, 2017, and 2018. We found that spiders shifted upwards in the canopy when woodlice were present, while grasshopper habitat domain was unchanged. Contrary to our predictions, the increased overlap in habitat domains did not consistently reduce grasshopper survival, despite a decrease in some years of the experiment. Our modeling work suggests that energy accounting can only explain the movement of spiders away from woodlice under opportunity costs or attack costs that are far higher than the available data suggest. We explore possible rationales for the change in spider behavior and provide insight into whether or not the outcome of this study can be generalized to other predators of similar ecological function.

## MATERIALS & METHODS

### Natural history

We conducted our study in old fields at the Yale-Myers Forest, which is a 3,213-ha forest in northeastern Connecticut (USA). Field experiments were approved by the Yale School Forests (project approval codes: BUCH15, SOM18, and SCH01). Old fields are abandoned agricultural fields supporting diverse perennial grasses and herbs, often dominated by goldenrods (e.g., *Solidago rugosa*). Within old fields, there are interconnected plant-based and detritus-based food chains. Our focal species in the plant-based food chain was the grasshopper *M. femurrubrum.* We chose *M. femurrubrum* because of its abundance in old fields at Yale-Myers Forest and its generalist diet. *M. femurrubrum* consumes both

grasses and forbs, and unlike many other old field grasshoppers, *M. femurrubrum* is rarely cannibalistic, meaning experimental populations of more than one individual can be tracked with causality to predator effects (*Schmitz, 2010*). *M. femurrubrum* is depredated by *P. mira,* a sit-and-wait nursery web spider and *Gladicosa gulosa*, a sit-and-pursue wolf spider. *P. mira* and *G. gulosa* can be found in the fields and forests, while other grasshopper predators including spiders *Phidippus clarus* and *Rabidosa rabida* are common in the field interior (*Schmitz, 2008*; *Schmitz et al., 2015*). *P. mira* and *M. femurrubrum* live at densities of approximately 1-m$^{-2}$ and 5-m$^{-2}$, respectively. The detritivorous woodlouse *O. asellus* was our focal species in the detritus-based food chain. It shares a habitat with the grasshopper and its predators along the edges of old fields. *O. asellus* lives at densities of approximately 30-m$^{-2}$ at the edge of old fields and at lower densities in the center of the field. The intersection of these seemingly separate communities, which is the focus of our study, may have important consequences for food web dynamics at the field-forest ecotone.

## Experimental design

We collected woodlice by hand from beneath wooden coverboards placed in forests and fields of the Yale-Myers Forest. When abundance was low, woodlice were also collected from beneath logs in the same area. Following collection, woodlice were stored in plastic bins with leaves, branches, and moistened cloth to provide cover and food. We collected nursery web spiders and third-instar grasshoppers from adjacent old fields using sweep nets and housed them in glass and plastic containers before transferring them into experimental cages. For a control treatment on predator hunting type, *G. gulosa* wolf spiders were collected from field margins by placing a bottomless trash can over a section of vegetation, disturbing the plants within, and catching any spiders that climbed up the sides of the can. All animals were collected from sites within a 16-km radius.

In the 2015 and 2017 studies, we constructed habitat domain cages using plastic boxes (l × w × h = 30 ×15×12-cm), metal fencing (100-cm tall), and window screen in which animals could be enclosed for behavioral observations (*Miller, Ament & Schmitz, 2014*). The cages were filled with sod cut from old fields using dibble sticks. Sod sections contained *S. rugosa* and other forbs, grasses, and detritus that had accumulated on the surface of the soil in densities representative of natural field conditions. Animals were placed into cages the evening before behavioral experiments to allow for sufficient acclimation.

For the 2015 and 2017 studies, we created three different experimental combinations of *P. mira, M. femurrubrum,* and *O. asellus*: (1) spider and grasshopper; (2) grasshopper and woodlouse; and (3) spider, grasshopper, and woodlouse. Treatments with woodlice contained six individuals to reflect densities in the field margins. Treatments with grasshoppers and spiders contained two and one individuals, respectively. This allowed for feasible yet sufficient observation of *M. femurrubrum* and *P. mira*. However, our experimental densities exceeded field densities because partial individuals could not be added (*Schmitz, 2004a*). To test whether or not the hunting mode of spider predators was an important factor, we included two additional treatments in 2017: (1) wolf spider (*G. gulosa*), grasshoppers, and woodlice; and (2) wolf spider and grasshoppers. Each treatment was

replicated five times in 2015 ($n_{Total} = 15$) and eight times in 2017 ($n_{Total} = 24$; excluding *G. gulosa* treatments where $n_{G.gulosa} = 4$).

In 2018, we added a factorial warming treatment to our experiment such that half of the cages were under heat lamps throughout behavioral observations. The treatment with only woodlice and grasshoppers was removed in 2018, since we had observed no effect of woodlice on grasshopper habitat domain after two years of experimentation. There were four treatments in 2018: (1) spider and grasshopper ambient; (2) spider and grasshopper warmed; (3) spider, grasshopper, and woodlouse ambient; and (4) spider, grasshopper, and woodlouse warmed. We placed HOBO temperature loggers in a subset of the cages to record the temperature. The heat lamps increased the cage temperature by 9 °C on average (Fig. S1). In 2018, we used larger, wooden bases for the cages (l × w × h = 30.5 × 40.6 × 6.4-cm) to improve our assessment of grasshopper survival and adjusted experimental population densities accordingly for a total of one spider and three grasshoppers. Each treatment was replicated seven times.

In all years, we followed established methodology for behavioral observations (*Miller, Ament & Schmitz, 2014*) by recording the spatial position (x, y, and z coordinates), behavior, and perch substrate used by each spider and grasshopper every 30 min between 07:00 and 19:00. In the first year of this study, we replicated behavioral observations on 5 August 2015 and 6 August 2015 for each cage. We verified that the results were quantitively similar and decided to run a single day of behavioral observations for each cage in the last two years of the experiment. These observations took place on 27 July 2017 and 9 August 2018. Behavioral observation days were clear and sunny in 2015 and 2018, while 27 July 2017 was sunny in the morning and overcast in the afternoon. Daily records from the nearest weather station at West Thompson Lake indicate that temperatures ranged from 13 to 28 °C, 8 to 23 °C, and 21 to 30 °C for the behavioral observation days in 2015, 2017, and 2018, respectively. For all behavioral observations, the cages were arranged in a randomized block design and placed on tables in open fields, under ambient light and temperature conditions unless experimentally warmed. Several undergraduate interns helped with data collection, so individual observers were allocated to blocks of cages so that observer error could be modeled as a function of blocks. Six hours of data from six cages in 2018 were lost from our database. We re-ran our analysis without these cages, because the habitat domains in these six cages were based on fewer data points. Since removing these cages did not alter our conclusions (c.f. Tables S1, S2), we retained the cages with fewer data points in the final analysis.

After completing the behavioral observations, we arrayed the blocks of cages in the same field and left them unmanipulated through September. Weather data from West Thompson Lake showed that 2017 was ∼2 °C cooler on average through August than 2015 or 2018 (Fig. S2). For the 2018 warming treatments, we wrapped the cages with plastic to raise their temperature using passive warming (*Barton & Schmitz, 2009*). The average temperature increase was 1 °C (Fig. S1). In September before arthropod-killing frosts, we opened the cages and collected all individuals to estimate survival. The harvest dates were 20 September 2015, 23 September 2017, and 22 September 2018.

## Statistical analysis

We analyzed our data using a Bayesian mixed modeling approach fit with Markov-Monte Carlo Chains. All models were fit using the R package *brms* version 2.8.0 (*Bürkner, 2017*). Our analysis occurred in two phases. First, we analyzed the height of spiders and grasshoppers in the canopy across treatments to determine whether the presence of woodlice and (in 2018 only) higher temperatures would influence their respective average heights. We averaged the height of grasshoppers and nursery web spiders in each cage and used average height as the dependent variable in our models. We used a multivariate model with grasshopper and nursery web spider height as dependent variables to account for any correlation between grasshopper and nursery web spider heights within a cage (Eq. 1).

$$L_h \sim MultiNormal(\mu_{hi}, \textstyle\sum)$$
$$\mu_{hi} = \alpha_i + \alpha_{i,Year[k]} + \alpha_{i,Block[j]|Year[k]} + \beta_{i,T} T_{hi} + \beta_{i,W} W_{hi} + \beta_{i,TW} T_{hi} W_{hi}$$
$$\textstyle\sum \sim LKJ(\sigma_i)$$
$$\alpha_{MEFE} \sim T_3(69, 10)$$
$$\alpha_{PIMI} \sim T_3(60, 28)$$
$$\sigma_{MEFE} \sim T_3(0, 10)$$
$$\sigma_{PIMI} \sim T_3(0, 28)$$
$$\beta_{MEFE,T} \sim Normal(-0.13, 16.1)$$
$$\beta_{PIMI,T} \sim Normal(-12.7, 9.4)$$
$$\beta_{i,W} \sim Normal(0, 100)$$
$$\beta_{i,TW} \sim Normal(0, 100)$$
$$\alpha_{MEFE,Year[k]} \sim T_3(0, 10)$$
$$\alpha_{PIMI,Year[k]} \sim T_3(0, 28)$$
$$\alpha_{MEFE,Block[j]|Year[k]} \sim T_3(0, 10)$$
$$\alpha_{PIMI,Block[j]|Year[k]} \sim T_3(0, 28)$$
$$i = \{MEFE, PIMI\}$$
$$j = \{1, 2, \ldots, 11\}$$
$$k = \{2015, 2017, 2018\}.$$

(1)

The above model predicts the effect of woodlouse presence (W) and heat lamps (T) on the heights of nursery web spiders and grasshoppers in the behavioral cages. The model produces estimates for the mean height $\mu_i$ and standard deviation $\sigma_i$ for each species, along with the woodlouse, temperature, and woodlouse × temperature interaction effects. We used a simpler model to analyze the wolf spider (*G. gulosa*) treatments testing for effects of predator hunting mode, because the wolf spiders were not observed often enough to fit a model with a joint spider and grasshopper distribution (Section S3).

The priors for our model are shown in Eq. 1 as distributions for intercepts (α), coefficients (β), and standard deviation (σ). We began with uninformative priors for all effects related to woodlice, since we had no *a priori* data. The variance of these priors used the default settings of the *brms* package. We constructed informative priors for the effect of temperature (i.e., heat lamps) on grasshopper and nursery web spider height using published data (*Barton & Schmitz, 2009*). Including these priors did not alter the qualitative

conclusions of our analysis (c.f. Tables S1, S3). The model included nested random effects for experimental block within year.

Next, we calculated the predicted attack rate of nursery web spiders on grasshoppers based on the overlap in their vertical distribution (*Northfield, Barton & Schmitz, 2017*; *Carroll et al., 2019*). We calculated a theoretical attack rate for each cage using the mean height and standard deviation across all the observations during the same day. We verified that the difference in the heights of grasshoppers and nursery web spiders was representative of the actual Euclidean distance between them in three dimensions using data collected in 2017 and 2018 ($R^2_{adj} = 0.72$, Fig. S3). We also examined the correlation between spider body size and grasshopper survival for data collected in 2015. The body size of nursery web spiders did not correlate with grasshopper survival (Fig. S4). We then tested whether grasshopper survival was reduced by increases in predicted attack rate across all the treatments and years (Eq. 2). We used a binomial distribution to model the effect of attack rate on the probability of grasshopper survival in each cage with block nested within year as a random effect.

$$L_i \sim Binomial(n_k, p_i)$$
$$p_i = \alpha + \alpha_{Year[k]} + \alpha_{Block[j]|Year[k]} + \beta_A A_i$$
$$\beta_A \sim Normal(0, 10)$$
$$\alpha \sim T_3(0, 10)$$
$$\alpha_{Year[k]} \sim T_3(0, 10)$$
$$\alpha_{Block[j]|Year[k]} \sim T_3(0, 10)$$
$$j = \{1, 2, \ldots, 11\}$$
$$k = \{2015, 2017, 2018\}.$$

(2)

We calculated the spatial overlap of grasshoppers and spiders with woodlice, even though the woodlice were not observed often during the day ($n = 4$ times in 2017). Based on their natural history, we assumed the woodlice spent most of the day belowground (*Hassall & Tuck, 2007*). At dawn and dusk, a larger portion of the woodlouse population forages aboveground in the leaf litter, while nursery web spiders and grasshoppers also shelter in the litter. Thus, woodlice most likely interacted with spiders and grasshoppers during the dawn and dusk, when behavioral observations were not possible because of low light levels. We assumed that nursery web spiders and grasshoppers with a habitat domain closer to the ground were more likely to interact with woodlice.

## Model simulations

We used three models to explore mechanisms that could explain shifts in habitat domain. The models are (1) a net energy gain model, (2) a novel combination of signal detection theory (*Abbott & Sherratt, 2013*) with habitat domain theory (*Northfield, Barton & Schmitz, 2017*), and (3) an individual-based simulation to test movement bias. For the first two models, we calculated energy gains and losses for spiders attacking grasshoppers and woodlice at different temperatures. The third model is a complementary model, while the first two are competing models based on the same fundamental framework. The parameters and their references are presented in Table 1.

**Table 1** **The parameters used in our theoretical models.** An asterisk indicates parameters estimated in-part or entirely from data measured on other species of spiders because data for *Pisaurina mira* were not available. Some parameters are used only in intermediate steps, see supplemental code for details.

| Parameter | Value | Reference |
|---|---|---|
| Grasshopper body energy content | 33.81 J | *Wiegert (1965)* |
| Spider handling time* | 20 min | *Samu (1993)* |
| Spider attack time | 0.5 min | Based on field observations |
| Temperature increase with canopy height during a summer day | 0.1 °C cm$^{-1}$ | *Bazzaz & Mezga (1973)* |
| Spider resting metabolic rate | Function of temperature | Derived from the respiration data reported in *Rosenblatt, Wyatt & Schmitz (2019)* |
| Spider active metabolic rate* | Function of temperature | Linear function of data reported in *Ford, 1977* and *Rosenblatt, Wyatt & Schmitz (2019)* |
| Respiration quotient | 0.7 | *Schmitz (2004b)* |
| Oxycalorific equivalent | 0.0200832 J ($\mu$l O$_2$)$^{-1}$ | *Ford (1977)* |
| Spider ability to distinguish grasshoppers and woodlice | 2.5 & Varied | *Abbott & Sherratt (2013)* |
| Probability of losing a prey item each attack opportunity | 0.25 & Varied | *Abbott & Sherratt (2013)* |
| Daily hunting period for spiders | 8 h | Set based on field observations |
| Spider attack success rate | 0.25 & Varied | Set based on field observations |
| Spider assimilation efficiency | 0.8 | *Moulder & Reichle (1972)* |
| Number of times per day that a spider encounters a prey | 0.8 | *Miller, Ament & Schmitz (2014)* and attack success rate |
| Spider movement probability without stimulus (i.e., an encounter) | 0.1 & 0.8 | Set to match empirical and simulation movement rates |
| Spider distance moved if movement occurs (mean $\pm$ standard deviation) | 10.5 $\pm$ 14.5 cm | Calculated from our behavioral data |

We used the net energy gain model to test the hypothesis that encounters with woodlice represent a significant cost to nursery web spiders. We assumed a spider always attacks a potential prey item and calculated the net energy gain for nursery web spiders hunting at different heights in the canopy. We calculated encounter probabilities using the vertical habitat distributions (*Northfield, Barton & Schmitz, 2017*). For model simulations, we increased the robustness of our height measurements for grasshoppers and spiders by adding two years of previously published data (*Miller, Ament & Schmitz, 2014*). Our habitat domains were normally distributed for spiders and grasshoppers, but fit to a gamma distribution for woodlice because their height distribution was highly skewed towards the ground (Fig. S6). We assumed equal densities of potentially interacting grasshoppers and woodlice, since the lower field density of grasshoppers relative to woodlice is approximately compensated by the 40–90% of the woodlouse population that shelters belowground even at dawn and dusk, and is therefore unable to encounter the spiders (*Hassall & Tuck, 2007*). This is a conservative assumption because increasing the population density of woodlice would increase the size of their effect.

The net energy gained by spiders from attacking grasshoppers and woodlice was positive and negative, respectively. We calculated the expected gain from attacking a grasshopper as the energy content of the grasshopper minus the energy lost to active metabolism during the attack and handling of the grasshopper. We also corrected for assimilation efficiency

and attack success rate by deducting the unassimilated energy and adding the cost of unsuccessful attacks (Table 1). The net energy gain of attacking a woodlouse was the cost of attacking only because we have no evidence to suggest nursery web spiders consume woodlice. We assumed that attacking grasshoppers and woodlice takes the same amount of time (Table 1). To account for temperature, we added the cost of increased nursery web spider metabolic rate when they were hunting or resting higher in the canopy (*Bazzaz & Mezga, 1973*; *Barton & Schmitz, 2009*; *Rosenblatt, Wyatt & Schmitz, 2019*). We plotted the expected net energy gain a nursery web spider should receive with and without woodlice for each perch height in the plant canopy.

We combined habitat domain theory with signal detection theory in the second model to evaluate whether or not nursery web spiders learn to discern woodlice after encountering them (*Abbott & Sherratt, 2013*; *Northfield, Barton & Schmitz, 2017*). Signal detection theory is a decision framework that can determine the best predation strategy given the density of possible prey, the density of non-prey items that look like prey, and the net cost of attacking both these organisms (*Abbott & Sherratt, 2013*). Here, we use signal detection theory to determine where spiders should perch and how often they should attack a potential prey item if both grasshoppers and woodlice are present. We relaxed the typical assumption in signal detection theory that encounter rates are determined solely by species density (as in *Abbott & Sherratt, 2013*), and instead calculated them as a weighted probability of habitat domain overlap (as in *Northfield, Barton & Schmitz, 2017*). We chose to use a signal detection model focused on the probability of losing a current attack opportunity, because we did not have data to parameterize spider opportunity cost (i.e., the value of watching for predators, searching for mates, etc.). We optimized spider canopy height and the number of times they would attempt an attack on an animal of questionable identity using the combined theory.

Finally, we ran a simple individual-based model to test the hypothesis that any differences in nursery web spider canopy height can be explained by spiders moving after a fruitless interaction with a woodlouse. Spiders were started at a random canopy height and moved from that height based on an empirically derived probability of movement and step size distribution (Table 1). Spiders moved after encountering woodlice. We ran two simulations: one with spiders that sit-and-wait for their prey (e.g., *P. mira*) and another where spiders more actively hunt for their prey (e.g., *G. gulosa* or *P. clarus*). We conducted a supplemental analysis where spiders also move after failing to capture a grasshopper. Encounter probabilities were calculated using the same habitat domain parameters as in the previous models modified by hourly encounter rates to make the model time-specific (Table 1). We present the results of 100 replicates of a 50-hour simulation with and without woodlice present.

## RESULTS

Our analysis suggests that spiders were more likely to move upwards in the canopy when woodlice were present ($\beta_{PIMI,W} = 16.52\ cm\,[4.36, 28.27], x\,[95\%\ CI]$). The expected movement of ~16-cm and the 95% confidence interval indicates that spiders moved

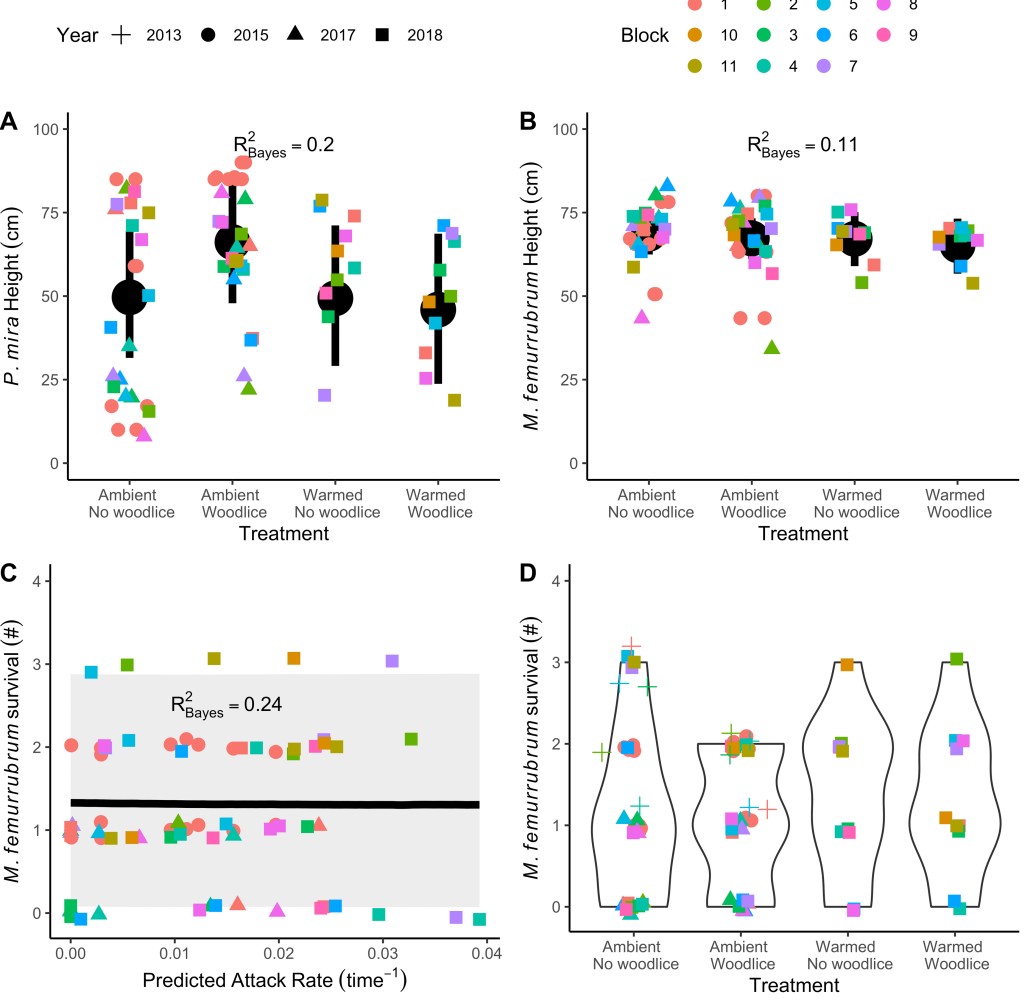

**Figure 1** **The impact of woodlice and temperature treatments on spider (A) and grasshopper (B) heights in the canopy and grasshopper survival (C–D).** Spiders (*Pisaurina mira*) move up in the canopy when woodlice (*Oniscus asellus*) are present assuming ambient temperatures (A), whereas grasshoppers (*Melanoplus femurrubrum*) do not change their height. Grasshopper survival is relatively consistent across trials, despite a difference in the 2013 data (C: +; data from 2013; *Buchkowski & Schmitz, 2015*). Grasshopper survival is not correlated with predicted spider attack rate, as calculated from the overlap of their respective space use. Blocks with the same number across years are not meaningfully related.

upwards on average (Table S1). Often, spiders moved from the mid-canopy to the top of the canopy (Fig. 1A). Temperature alone did not cause spiders to move lower into the canopy in contrast to previous studies ($\beta_{PIMI,T} = -0.45 [-13.65, 12.64]$). When spiders experienced both warming and woodlouse presence, the interaction nullified the effect of woodlice alone ($\beta_{PIMI,TW} = -19.28 [-40.65, 1.45]$), suggesting temperature stress was more important than woodlice to spiders (Fig. 1A). Grasshoppers did not change their position in the canopy in response to woodlouse ($\beta_{MEFE,W} = -1.44 [-6.81, 3.83]$) or temperature treatments ($\beta_{MEFE,T} = -1.58 [-8.94, 5.92]$).

The movement of nursery web spiders higher into the canopy increased the overlap between grasshopper and spider habitat domains (Figs. 1A–1B). However, in contrast to our predictions, we found that increases in the theoretical spider predation rate did not actually reduce grasshopper survival across our entire dataset ($\beta_A = -0.54\,[-17.41, 16.01]$; Fig. 1C; Table S4). Grasshopper survival was lower in spider × woodlouse cages in 2013 and 2017 (Fig. 1D), but this trend did not occur in 2015 or 2018. The 2013 data were taken from an earlier publication to extend our dataset (*Buchkowski & Schmitz, 2015*).

Wolf spiders (*G. gulosa*) remained hidden in the leaf litter throughout most of our observations, and we only observed two individuals for a total of three unique positions in the canopy. Consequently, we could not model the joint grasshopper and wolf spider habitat domain. The wolf spiders caused the grasshoppers to move upwards in the canopy regardless of whether woodlice were present ($\beta_{GW} = 15.85\,[3.30, 28.09]$) or absent ($\beta_G = 13.72\,[2.19, 24.94]$; Fig. S5; Table S5). Grasshopper survival was not altered by the presence of wolf spiders (Fig. S5). We did not pursue this line of investigation further, because there was no woodlouse × wolf spider interaction and wolf spiders were not observable.

Our model simulations show that nursery web spider behavior is more likely driven by movement away from woodlice rather than movement towards grasshoppers (Fig. 2). Calculating the energy balance demonstrated that attacking woodlice has a negligible effect on spiders, even if they attack every woodlouse they encounter (Fig. 2A). These results only change if the cost of attacking woodlice is higher by three orders of magnitude (Fig. S7A) and do not change if spider attack success is increased (Fig. S7B).

Our combined model of signal detection theory with habitat domain predicted that spiders should employ an "always attack" strategy when woodlice are present (Fig. S8). An "always attack" strategy means that spiders should attack both woodlice and grasshoppers, because the cost of missing a grasshopper outweighs the energy wasted attacking a woodlouse. These models predict that spiders should only shift their habitat position in the canopy when they (1) have great difficulty distinguishing between woodlice and grasshoppers, or (2) face a large risk of losing that prey item, or (3) face a cost of attacking woodlice three orders of magnitude higher than we expected it to be (Fig. S8). Extreme cases are also the only situations where spiders should make multiple attack attempts in order to learn the difference between woodlice and grasshoppers.

The individual-based model simulations replicate our empirical observations of sit-and-wait spiders shifting upwards when woodlice are present (c.f. Fig. 1A, 2B). The sit-and-wait simulations replicate the empirical probability of nursery web spider movement (sim. = 0.092, emp. = 0.104) and the correct order of magnitude for empirical average distance moved by the spiders (sim. = 1.21 ± 1.47 cm, emp. = 2.35 ± 9.49 cm). The qualitative results are the same if spiders move after unsuccessful attacks against a grasshopper as well (Fig. S9A). The difference in average spider height caused by the presence of woodlice disappeared when we increased the spider movement probability from 0.1 to 0.8 to replicate an active hunting predator (Fig. 2C). The biased movement away from woodlice only appears to matter if the spider predator has an otherwise low probability of moving. Our theoretical analysis suggests that the woodlouse effect on spider habitat domain is

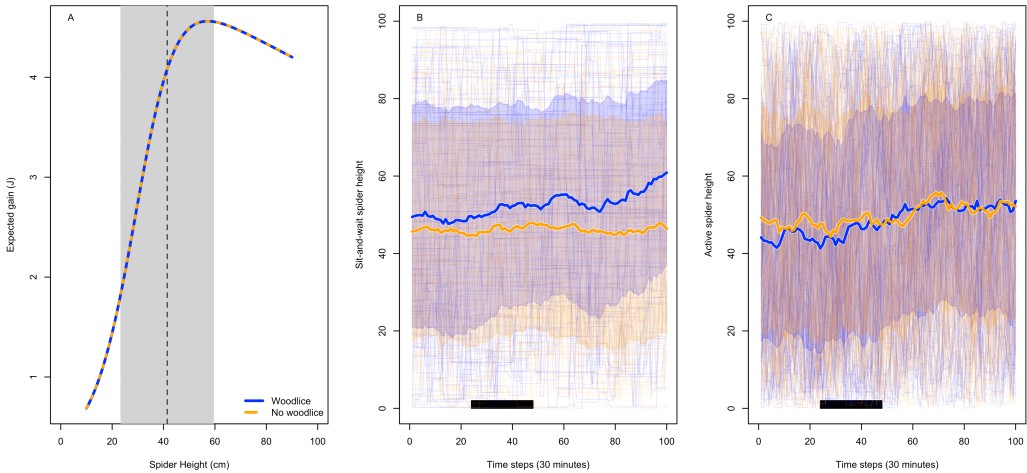

**Figure 2  Models of spider foraging height based on maximizing net energy gain and avoiding woodlice.** A simulation of expected energy gain of spiders hunting grasshoppers, based on their height in the canopy, in cages with and without woodlice (A). Woodlice presence does not change the expected energy gain when only the respiration costs of spider attack are considered. The dashed line and grey box indicate the empirical spider mean height and ±1 standard deviation combining our data with *Miller, Ament & Schmitz (2014)*. Woodlice presence increases the average height of sit-and-wait (B) but not active hunting (C) spiders in an individual-based simulation where spiders move hunting perches after an encounter with woodlice. Thick lines and shading show mean ± 1 standard deviation, thin lines show 100 individual trajectories, and the black bar shows the times when our empirical behavioral observations occurred.

more consistent with spiders switching perches after encountering a woodlouse, rather than because they are maximizing net energy intake.

## DISCUSSION

Our results indicate that the nursery web spider predator *P. mira* shifts its habitat domain upwards in the canopy when the woodlouse *O. asellus* is present. Woodlice appear to be a poor prey item for the spider due to the low nutritional value of woodlice and the spider's inability to handle the morphology of a large, armored arthropod (*Vizueta et al., 2019*). We can conclude that their interaction likely links two disparate food chains via a trait-mediated effect.

In general, sit-and-wait predators like *P. mira* depend on detecting movement to capture nearby and unsuspecting prey (*Lawrence, 1985*; *Gall & Fernández-Juricic, 2009*). Therefore, a plausible explanation for the behavioral change was that the movement of woodlice disturbs or distracts the spider enough to induce it to move away. The sit-and-wait hunting strategy of the spider is successful only under low energetic expenditure (*Schmitz, 2006*), so reacting to false alarms could come at a high cost. We predicted that the spider avoids the false alarms of woodlouse movement by shifting higher into the canopy (*Green & Swets, 1966*; *Getty & Krebs, 1985*). However, our analysis of spider energetic balance did not support this hypothesis. Using a novel combination of signal-detection theory and habitat domain theory, we show that if a spider is optimizing energy intake, it should

attack everything because the cost of a short burst of activity (~4.3 mJ) is small relative to the expected payoff of attacking a grasshopper (~6.8 J). Consequently, spiders do not benefit much from 'testing' multiple items to learn to distinguish between woodlice and grasshoppers (*Abbott & Sherratt, 2013*). Attacking costs would need to be three orders of magnitude higher to alter the optimal spider strategy (Fig. S8).

An alternative explanation is that nursery web spiders change their perches after incorrectly attacking a woodlouse. *P. mira* spend most of the day in a single location (our data, *Miller, Ament & Schmitz, 2014*; *Rosenblatt, Wyatt & Schmitz, 2019*), so unsuccessful attacks on woodlice can be infrequent but would still induce a change in the median canopy height of the spider. Furthermore, interactions with woodlice occur almost exclusively near the ground where the only direction to move is upwards. A similar response to an unsuccessful attack on a grasshopper would not have the same effect, because grasshoppers occupy the middle to upper canopy (c.f. Fig. 2B, Fig. S9). However, when the spider's baseline movement rate is increased, as would be the case for an active hunting predator, the effect of woodlice is washed out. Our individual-based model suggests that this small bias in movement probability can lead to an increased height only for spiders that do not move often.

We might question why *P. mira* would ever perch in the lower canopy when grasshoppers are often in the middle to upper canopy. Our model and previous empirical data suggest that thermal stress higher in the canopy may drive *P. mira* to perch in the lower canopy (*Barton & Schmitz, 2009*). This is supported by our data showing that *P. mira* remained in the lower canopy under warming conditions, regardless of the presence of woodlice. Another explanation is that nursery web spiders also feed on other old field arthropods and likely respond to the habitat domain of these prey. For example, the intraguild prey species *P. clarus* occupies the middle canopy, close to the perches *P. mira* selected in woodlouse-free cages (*Miller, Ament & Schmitz, 2014*). Future studies could explore whether the movement of *P. mira* upwards in the canopy reduces predation on other arthropod prey.

In treatments combining woodlice and warming, the nursery web spider *P. mira* did not move upward. The spider may be too heat sensitive to move upwards in the canopy in response to woodlice (*Barton & Schmitz, 2009*). Under this hypothesis, thermal stress is the binding constraint, but relaxing thermal stress allows the spiders to choose a foraging perch away from woodlice. Therefore, global climate change trends may influence the relationship between these animals. The effects of woodlice on spiders, along with any subsequent effects on ecosystem properties, would be mitigated or eliminated if nursery web spiders no longer move upwards (*Buchkowski & Schmitz, 2015*). We must acknowledge that warming may also influence the ecosystem in ways that our short-term study cannot predict. One possibility is that warming temperatures will increase grasshopper growth rates (*Coxwell & Bock, 1995*), reducing the capture success for predatory spiders. This could shift spider diets toward other organisms, subsequently altering grasshopper diets and nitrogen cycling.

The effect of woodlice on grasshopper survival was inconsistent across the three experimental years. One explanation is that the nursery web spider has a negligible impact on grasshopper survival even if encounter rates are increased (*Schmitz, 2010*)

because their attack success rate is low. Two lines of evidence support this conclusion. First, our predicted spider attack rates do not correlate with grasshopper survival. Since we know that the spider does not reduce grasshopper populations substantially, it is probable that their attack success is low enough to negate the effects of a higher encounter rate (*Schmitz, 2006*). The second line of evidence is correlative and provides one explanation for why grasshopper survival was only reduced in two years: 2013 and 2017. The largest effect of woodlice on grasshopper survival occurred in 2013, when the cage experiment started earlier in the season (Fig. S3). While we always stocked third-instar grasshoppers into our behavioral experiments, there may have been size variation within the instars at stocking time that could, on average, select for smaller individuals earlier in the season. Smaller grasshoppers are more susceptible to spider predation (*Brose, 2010*). The larger effects that we observed in 2013 and 2017 may have been caused by rapid consumption of smaller grasshoppers in the first week of the experiment. Another factor may have been the differences in annual climate. The summer of 2013 had the warmest July and coolest August of any study year. It is possible that cooler August temperatures kept grasshoppers small in 2013 and 2017, increasing spider predation success (*Coxwell & Bock, 1995*). A climate explanation is also consistent with the interaction effect of woodlice and warming on spider height in 2018, because the spider's upward shift was smaller in warmer years (c.f. Fig. 1A; Fig. S3). Overall, our results suggest that spider attack success rate is low relative to encounter rates, mitigating the differences in grasshopper survival in all but the most ideal circumstances.

Measuring parameters such as attack success requires controlled experimental conditions where reliable data can be collected. Our small cages balance two features: (1) maintaining key environmental conditions such as plant canopy structure and temperature and (2) keeping the same animals contained and observable over an entire day. Larger cages closer to 1-m$^3$ make it impossible to observe the animals when they shelter in the center (2015, R. Buchkowski). The controlled environment was especially important for factorially manipulating woodlice, because they must be disturbed by removing cover objects to observe them in their natural habitat (*Hassall & Tuck, 2007*; our study). We were able to parameterize our models of spider, grasshopper, and woodlouse interactions because our simplified cage environments allowed us to manipulate animals and observe interactions.

Our results demonstrate that behavioral interactions can link plant-based and detritus-based food chains even without cross-chain feeding. Specifically, we demonstrate how the behavior of a spider in the plant-based food chain is altered by the presence of a detritivore. We did not observe direct interactions between nursery web spiders and woodlice in our cages. Further attempts to observe these interactions would provide a definitive test of our hypothesis that "failed attacks" instigate spider movement. Although grasshopper survival was unaffected overall, woodlice may be perturbing the entire food web by impacting the the location of the nursery web spider in the canopy. Future studies involving prey species that are consumed by nursery web spiders in the mid-canopy, such as the intraguild prey *P. clarus*, would help test our model (*Barton & Schmitz, 2009*). One intriguing possibility is that the shift in habitat domain is a functional trait of sit-and-wait predators, whose location in the canopy changes less often than active hunting predators (*Schmitz, 2010*).

This hypothesis is supported by the behavior of the individual-based model. Data from an active hunting spider that can be observed in the canopy, such as *P. clarus*, would provide a direct test of our hypothesis that the response of nursery web spiders to woodlice is linked to their sit-and-wait strategy.

## CONCLUSIONS

We provide a new modeling approach that combines signal-detection theory and habitat domain theory to help predict when predators hunting in a spatially structured environment make energy trade-offs between true and false prey items. Using our empirical data from old fields, simulations demonstrated that spider predators were not sensitive to the energetic trade-off. Instead, our modelling work supported the alternative hypotheses that the shift in habitat domain resulted from the movement of nursery web spiders away from an undesirable interaction. Future research into sit-and-wait predators could use this theory to elucidate how habitat domain and species interactions might link seemingly disparate food chains and shift with changes in local climate.

## ACKNOWLEDGEMENTS

We would like to thank the many interns who helped collect behavioral data and the Yale School Forests for providing the facilities necessary to conduct our research. We thank Oswald Schmitz and Max Lambert for providing invaluable feedback as we designed our experiments and analysis.

### Funding

The work was funded by the Schiff Fund and the Yale School of Forestry & Environmental Studies. We received in kind contributions from the Yale School Forests. Robert Buchkowski was supported by the Natural Sciences and Engineering Research Council of Canada (PGSD3-454293-2014). The funders had no role in study design, data collection and analysis, decision to publish, or preparation of the manuscript.

### Grant Disclosures

The following grant information was disclosed by the authors:
Schiff Fund and the Yale School of Forestry & Environmental Studies.
Natural Sciences and Engineering Research Council of Canada: PGSD3-454293-2014.

### Competing Interests

The authors declare there are no competing interests.

### Author Contributions

- Stefanie M. Guiliano and Cerina M. Karr conceived and designed the experiments, performed the experiments, authored or reviewed drafts of the paper, and approved the final draft.

- Nathalie R. Sommer and Robert W. Buchkowski conceived and designed the experiments, performed the experiments, analyzed the data, prepared figures and/or tables, authored or reviewed drafts of the paper, and approved the final draft.

## Field Study Permissions

The following information was supplied relating to field study approvals (i.e., approving body and any reference numbers):

Field experiments were approved by the Yale School Forests (project approval numbers: BUCH15, SOM18, and SCH01).

## Data Availability

Data and the model code is available at GitHub: https://github.com/robertwbuchkowski/woodlice-and-nursery-web-spiders.

## Supplemental Information

Supplemental information for this article can be found online at http://dx.doi.org/10.7717/peerj.9184#supplemental-information.

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
