# Peer review of "Woodlice change the habitat use of spiders in a different food chain"

_PeerJ, doi:10.7717/peerj.9184_

## Round 0.1 · original submission · Minor Revisions

I share the reviewers' view that this manuscript is a generally sound piece of work in need of minor revision. The reviewers make many constructive suggestions for improving the presentation of the existing information, and I share their opinions. In addition, I agree that more methodological detail needs to be added to the Methods section. Please address each comment in your revision.

Reviewer 1 ·

Basic reporting

Language is generally clear, but there are some vague statements that need to be clarified as mentioned below in the comments section. Also the MS could use a good read over and edit for typos. I started pointing them out but there are quite a few.

Backbone structure of the paper is good but it is strange that results from the data outlined in the results are presented in the introduction. I realize this it done to set up the 3 models created to explore the first results presented but it makes it quite confusing to follow. I think they can more broadly present implication of the question without stating the results in the introduction (i.e. we present follow-up models testing X). This problem is reflected to some degree in the structure of the abstract as well (see comments)

The three models are presented as alternatives but the third model really seems like a complementary rather than a competing model and their interpretation seems to support this viewpoint as well. Rewriting throughout to clarify the relationship between the models and explicitly state how they are being used would help immensely.

Good job on archived data and code! It would be nice if the Readme file in github did a better job of referencing which code and data files specific figures and analyses in the paper and the pipeline that generated them.

Experimental design

The hypothesis being tested in some instances is vague. For example, in the abstract, the methods section states that "hypotheses" are tested but only one is explicitly mentioned in the background section above (this may be due to a confusing writing style). Further, I found that the paper would benefit from more specificity about which hypothesis were being examined by each of the three models.

The statistical design is well communicated and rigorous- the experimental field design requires more explanation before publication (see comments). Additionally, I learned new and relevant information about the field set-up in the discussion (i.e. the instar of grasshopper observed was standardized) that needs to be presented in the methods instead.

Validity of the findings

There are some unsupported conclusions. For example, on line 431 they claim that they demonstrate feed-backs between plant and detrital food-webs. From the data here a uni-directional impact is shown ( woodlouse makes spider move= detrital --->plant) I don't see any effect back to the detrital system. On line 437 I am not sure that this paper shows that sit and wait predators have that much of a functional impact specifically. This paper demonstrates that the spiders move, but shows no impact on grasshopper predation rate and no data showing previously documented effects on soil N in the absence of effects on predation.

Additional comments

Guiliano et al leverage 4 years of data to pursue the mechanism behind a surprising result from a previous study. This work (by the last author) found that the presence of spider predators results in changes in soil nitrogen due to the impact of predator on grasshopper behavior. The presence of woodlouse individuals in experimental mesocosms disrupts this effect leading to higher predation rates of grasshoppers by spiders which is coupled with loss of impact of spiders on soil nitrogen. This impact on ecosystem processes is not examined in this paper. The paper takes this result as a jumping off point and considers why spider predation rates may go up in the presence of wood lice. The authors hypothesize that it may be due to either spiders moving higher in the canopy or grasshoppers moving lower resulting in more encounters. They gather behavioral observations from 1 day each (I think) of habitat domain from four summers (2013, 2015, 2017, 2018) where spiders, grasshoppers and woodlice were manipulated. They let the mesocosms rest for the summer and then recover the individuals at the end of the season. They find that spiders move up in the canopy when woodlouse are present rather than grasshoppers shifting down. However, they are not able to replicate across years the result of higher predation rates of grasshoppers by spiders in the presence of woodlouse. In 2018 they add a “warmed treatment” to test whether spiders moving up in response to woodlice is counterbalanced by warming that has been shown to move spiders downward. They demonstrate that this does occur. They then run three models to figure out what the best explanation for the spider movement is. One is a net energy model, one a combination of signal detection theory and habitat domain theory incorporating encounter rate based on observed domain overlap, and one is an individual-based model of spiders moving away from encountering a woodlouse. They find that the best model reflects that spiders move away from encounters with woodlice rather than move toward grasshoppers to maximize net energy gain. They do not uncover as far as I can tell the reason WHY the spiders move away because no cost is detected for interacting with the woodlouse.

In general, the study design seems appropriate along with the analytic approach which nicely deals with multiple sample years and unbalanced experiments across years. The models are certainly a value-add to the paper and there has clearly been much effort to parameterize them properly. However, I found the framing and layout of the paper fairly difficult to follow with a number of logical jumps that need to be clarified for the reader. In addition, there are quite a few details on the experimental approach that were missing and are necessary before the paper is suitable for publication. The authors do have their data and code available.

Section specific comments:

Title: I am hesitant say that the title is strictly true for the data presented in this paper. Certainly the paper this one is based on does that, but here what is being demonstrated is 1) an impact of the woodlouse presence on a spider which causes a domain shift. However, no impact is documented of an impact of this shift on either the detrital or plant food chain as there is no effect of this shift on grasshopper survival and no data on soil processes presented.

Abstract: In general, this needs to be proofed for wood choice and specificity. For example, in the background it is unclear whether the “new hypothesis” is being tested in this study or a previous one. In the abstract background it is assumed that the spider moves but the intro does a nice job of setting up the option of either the spider or the grasshopper moving which this paper does nicely test! I would again use that framing.

Ln 40: Is maximizing energy gain the same as shifting toward the grasshopper? what does this mean biologically?
Ln 42: the biotic factor of alternative prey comes out of nowhere in the abstract and it isn’t mentioned much in the rest of the paper.

Ln 54: Insert “consumer” in front of species
Ln 54: Plant and detrital-based food chains are referenced throughout but are not really defined. Since in some sense of the term detrital food chains are also “plant-based” what is the difference? Is it that detrital is microbial necromass- based? It it really living vs dead plant tissue based? Clarify this early

Ln 61- missing “a”

Ln64: how does this link break? a directional change would be useful.
Ln66: If the woodlouse is a common part of this system then presumably the original mesocosm studies that detected the impact of spiders on nitrogen also had “hidden” woodlouse treatments. Wouldn’t they have broken the link in those other experiments as well?

Ln 72- How does lower grasshopper survival relate to the soil N result?

Ln 80- this line seems to contradict the feeding trial data. reconcile?

Ln81- insert “in response to woodlice”

Ln 93- This assumes that the grasshopper doesn’t also move in response to warming

Ln 97- vague. explain what the concept means in practice

Ln 115- why is this behavior unexpected?

Ln 117- Outline the hypotheses tested by the 3 models in the intro and provide background

Ln 94- this is a convoluted way of saying “whether spiders or grasshoppers respond”


Experimental details missing:

When were the behavioral observations started each year?

How many days of observations were completed each year?

How many degrees did the warming treatments raise the temperature?

Adult or nymphal grasshoppers stocked? Which instar if the latter?

Were spider sizes similar and/or was sex a potential determinant of predation? Do only some sizes of spiders eat grasshoppers?

Other guilds of spiders are reported being in the experiment for controls and then the data is discarded (although it is in the supplement). A short statement of what it shows and why it isn’t considered further seems appropriate?

What are the field densities of each player? They are referenced in the discussion but methods seems more appropriate

Ln 169- do these reflect field densities or margin densities

Ln 204- Is the date of September the same for all years of the experiement?

Ln 273- But I thought the woodlice and grasshoppers/spiders come in contact with each other in the early morning on the ground?

Ln288- define signal detection theory before relaxing assumptions

Ln 333- what is being always attacked? woodlice, grasshoppers or both?

Ln 346- why would they move away if there is no cost to attack?

Ln 369- link to the analysis name in the code

Ln 371- is “inappropriate” the best word here?

Ln 410- in your model attack success is 25%- this doesn’t seem that low…

Ln 431- this feedback seems unidirectional to me

Ln 445- change “a bias” to biased.

Reviewer 2 ·

Basic reporting

no comment; refer to general comments

Experimental design

no comment; refer to general comments

Validity of the findings

no comment; refer to general comments

Additional comments

This is an ambitious study that incorporates carefully designed cage experiments, sophisticated statistical modeling, and mathematical modeling of behavioral interactions to evaluate which of two basic explanations for a previously documented shift in foraging position of a generalist predator is most likely. The addition of the impact of global warming is somewhat of a stretch conceptually, but is so clearly presented that it should be retained. The manuscript is extremely well written and the supporting material is thorough. These two points are critical strengths, because the major contribution of this study will not be its findings, but rather its broadly comprehensive research design. Thus, readers who are intrigued by their synthetic approach, or have a more narrow interest, such as learning about Bayesian modeling, or the “novel” combination of the “habitat domain theory” and “signal detection theory,” will be able to consult the references and the on-line code and data files to explore the details for themselves. Those who are skeptical of their approaches will also be able to make an in-depth evaluation.
I am concerned about the degree to which these data may have been over-modelled, and the danger of generalizing from the results of modeling a highly simplified cage environment to being better able to “anticipate changes in ecosystem dynamics” (last sentence of abstract); however, these reservations reflect a personal philosophical point of view. Readers will be able to evaluate for themselves the validity of the approach, and most importantly, should be able to re-construct all the details of the statistical and dynamic modeling is they should so desire.
My only major suggestions are that (1) the authors spend considerably more time, in the final part of the Discussion, pointing out which specific behavioral interactions should be studied in more detail (such as the actual behavior of the predator when encountering the detritivore – just one example) as a means to demonstrate the strength, or weakness, of their models; and (2) defend, in the Discussion, the value of very simplified cage experiments as a means to estimate parameters needed for more complex, and hence realistic, dynamic models that may someday actually describe basic features of the dynamics of this old field ecosystem. The manuscript is OK without these changes, but I feel that adding them will increase the readership considerably by convincing skeptics of the power of the research approach this manuscript espouses.

Some more-minor suggestions:
Lines
Title Rather bland. It doesn’t capture the breadth or major contributions of the research; in fact, this general conclusion is already known for this system.
49-50 Overly broad statement of how two chains can be linked; for example, consumers can be linked indirectly because their two food chains are joined by a shared top generalist predator, a linkage that is not a trait-mediated indirect interaction.
65 link BRINGS in the
80 does “carnivorous” = “predaceous” in this case, or are they scavengers?
114 “significant” is either too broad or it implies “statistical significance” – both usages should be avoided --- especially for Bayesians!
128-32 These facts suggest that your cages are not caricatures of the old field community, but instead, of the narrow ecotone between forest and field.
138 THE spider . ..
159 specify basal and height dimensions
167 “communities” is a bit of a stretch: six detritivores, two hoppers and a spider do not a community make; strike out “communities using” and go straight to “combinations”
172 Give actual range of field densities; “slightly” is too slippery.
177-8 Why not consider these data? Do they not support the other conclusions? Are they too few to make generalizations possible? If the latter, they probably should not be included even in the supplemental material. If the latter is not true, they should be presented and discussed, since they describe the results of a type of “replicated” treatment.
194-5 These were cages placed on tables in the field – which is fine, but these cages do not incorporate the complexity of the old field community (even though sod was used), and thus should not be termed “mesocosms” in the Abstract or throughout the manuscript. This term is unfortunately vague, but the suffix “cosm” implies a somewhat natural world, which this carefully contrived collection of a few individuals in a cage is not. Again, a matter of opinion, but any usage that can reduce skepticism in the reader’s mind will increase the likelihood the reader will be open to what your study has to offer.
197-8 Excellent.
199-02 Give numbers in the Suppl. Materials
203-09 More details are needed here and throughout the Methods section: dates each year so that collecting period and elapsed time between events are clear: temperatures under the heat lamps, and plastic-wrapped cages, compared to controls; instars of hoppers added, and when they were added; differences between years in average temp and rainfall during the experiment. Some of these points are brought up in the Discussion, but most of them should be in the Methods section.
216 clarify “began with the average height”, because this statement appears to be contradicted by the following phrase, which implies that the 30-minute observations were the raw data points, which is the case, correct? This distinction is critical to understanding the basic structure of your mixed-model.
247-53 Lack of actual observations between woodlice and spiders is a major weakness of the experiment – interactions at dawn, dusk and during the night should have been made. This absence of direct observational data does not invalidate your modeling approaches, but it would seem to weaken them considerably. This fact is the basis of my general recommendation that in the Discussion you address future empirical work that would strengthen confidence in your modeling results.
312-3 “does not cross zero” – please remove this (sole) reference to NHST!! Your data presentation is simple and elegant, with parameter estimates and 95% Credible Intervals – this approach is excellent, so don’t tarnish it with statements (implied just as bad as explicit) of “statistical significance.”
373 change ; to ,
388 arthropods; however, this
437-8 “sweeping” is a tad over the top. In fact, the whole sentence 436-8 is overly broad, and it really does not encapsulate the major contributions of your research.
440 This line begins to better describe your major contributions – refer to my earlier general comments on how to strengthen the impact of your study.
446-8 Ditto, i.e. comments on line 440 apply here also.
Fig. 1C I find these colors difficult to distinguish against the dark-gray background.

---

## Round 0.2 · Minor Revisions

Your revisions have greatly improved the manuscript. Nice work! I appreciate your attention to each reviewer comment and the care with which you responded. It's nearly ready, but I caught a small number of minor issues in my careful reading. My comments are on an annotated version of the manuscript (attached). Please fix these with minor revisions and send back the revised version when you can.

---

## Round 0.3 · accepted · Accept

I appreciate your conscientious efforts to revise and improve your manuscript. I think it will make a very nice contribution to the field.